

# Scenarios of availability of water due to overexploitation of the aquifer in the basin of Laguna de Santiaguillo, Durango, Mexico

María de Lourdes Corral-Bermudez[1,*], Eduardo Sánchez-Ortiz[1,*], Dioselina Álvarez-Bernal[2], Martín Omar Gutiérrez-Montenegro[1] and Erika Cassio-Madrazo[1]

[1] Centro Interdisciplinario de Investigación para el Desarrollo Integral Regional Unidad Durango, Instituto Politécnico Nacional, Durango, Durango, México
[2] Centro Interdisciplinario de Investigación para el Desarrollo Integral Regional Unidad Michoacán, Instituto Politécnico Nacional, Jiquilpan, Michoacán, México
* These authors contributed equally to this work.

## ABSTRACT

**Background:** The importance of water to life is unquestionable. Most of the fresh water we use for daily activities comes from the aquifers, which in many cases due to misuse are overexploited and at risk. This article studies the aquifer that appeared in Laguna de Santiaguillo basin; it should be noted that the most important economic activity in the basin is agriculture.
**Methods:** By analyzing vector and demographic information, using GIS and with some field trips, the impact and risk on the level of disposition in every micro-basin that forms the basin were determined.
**Results:** The different modeling scenarios demonstrate that the basin and sub-basins that conform Santiaguillo are overexploited.
**Discussion:** The volumes concessioned are of such magnitude that they generate a condition of vulnerability to the activities of the basin and sustain the overexploitation conditions of the aquifer.

# INTRODUCTION

Water is an indispensable resource to almost all life in the world; one of the characteristics of this element is its potentiality to be renewable. The availability of superficial fresh water is calculated between 9,000 and 14,000 $km^3$, and just 4,200 $km^3$ are available for human consumption (*Cantú-Martínez, 2012*). A total of 2% of all fresh water is frozen in the polar ice caps. On the other hand, groundwater is stored at a depth of 1,000 m, representing 0.5% availability. This means a greater availability than that in rivers and the sea (*Fernández Cirelli & Du Mortier, 2005*).

The availability of fresh water in Mexico is about 475 $km^3$, 63 $km^3$ of this water is in the underground, and the other 412 $km^3$ is surface water (*Domínguez Alonso, 2011*).

Corresponding author
María de Lourdes Corral-Bermudez,
mcorralb1000@alumno.ipn.mx

The annual average rainfall in Mexico is of a volume of 1,511 km$^3$, which classifies Mexico as a country with low water availability (*FEA-CEMDA, 2006*).

The term average annual availability of surface water in a hydrological basin in its natural state (without water use) is the limit of the annual average volume of water that can be granted or assigned in a sustainable manner. On the other hand, in a basin in which some use is already made, the average annual availability of surface water is the portion of that limit that continues to be available for concessioning (*Silva-Hidalgo et al., 2013*).

The concept of availability of the NOM-011-CNA-2000 has a juridical-administrative or regulatory meaning, since it has the purpose of defining the existence of volumes of water susceptible to being concessioned in a basin, without affecting the volumes previously granted (*Silva-Hidalgo et al., 2013*).

The availability of water is threatened due to some economic activities of great importance such as industry and agriculture which require large volumes of fresh water to be carried out. As a result, aquifers suffer overexploitation.

As Custodio states: "*an aquifer is considered to be overexploited, or at risk of overexploitation, when the sustainability of existence is an immediate threat as a consequence of abstraction being grater, or very close to, the annual mean volume of renewable resources, or when abstraction may produce a serious water-quality deterioration*" (*Custodio, 2002*).

At a global level, aquifers present a serious threat due to both pollution and overexploitation caused by urbanization, industrial development, agricultural activities, and mining ventures (*Foster et al., 2002*).

In Mexico, the water administration has caused that 106 of the 653 aquifers be overexploited and some of them show deterioration in the quality of groundwater, data expressed by experts during the Second National Colloquium "Groundwater in Mexico" carried out in the senate (*Reforma, México D.F. México, Cervantes, 2015*).

Groundwater is one of the most important natural resources, the overexploitation of aquifers generates that the salinity of groundwater increases and continues to threaten the viability of agriculture (*Khezzani & Bouchemal, 2018*).

When an aquifer is declared overexploited, there are some actions and factors that should be considered, some of them are the legacy and to inform the population of the conditions in which the aquifer is found, as well as the remedial actions that can be carried out for the recovery and recharge of the aquifer.

All the rules, issues and procedures involve peasants, sellers and groups with agriculture interests as well as politicians, legislators, and judges. They control the access to the means of production and establish the economic and social interests in relation to the natural, technological, and human resources. But they also install the legitimate forms to transfer the resources either by allocation, interchange, sale, or heritage (*Moreyra et al., 2012*).

The capitalist-neoliberal model of food production is incapable of solving the problems of food security. The control, form and conditions under which unregulated capital operates in the world diet, activate the conditions for increasing insecurity, both in terms of food quantity and quality (*Altieri, 1999*).
Modern agriculture, with intensive abuse of industrial chemicals and bad distribution of water, has a negative impact on the environmental and human health. Soils, lakes, rivers, and groundwater suffer the impact of pollution (*Martínez, 2009*).

There are some studies that demonstrate the risk of an overexploited aquifer. An example of this is the case of the aquifer of Disi in Jordan. The study reveals that the environmental consequences of groundwater extractions generate dropping water levels, mobilization of some salty water bodies in geologic units connected vertically with Disi aquifer such as Khureim Formation and irrigation return flow waters with their high salt contents, biocides, and fertilizers concentrations (*Salameh, Alraggad & Tarawneh, 2014*).

## The aquifers in Mexico

Mexico is divided into 653 aquifers, of which 448 national aquifers are in a condition of availability, as of 2016, 105 overexploited aquifers were reported in the country, considering overexploitation based on the extraction/recharge ratio (*Mexican National Commission of Water (CONAGUA), 2017b*) (http://sina.conagua.gob.mx/publicaciones/EAM_2017.pdf).

The north and center of Mexico, have the largest amount of overexploited aquifers, since there is more population but less surface water, so that more groundwater is used (*Díaz et al., 2013*).

A poor use of resources places the population in an imminent risk. It is important to consider that the risk is the vulnerability that is conjugated with the threat, taking into account that nature is not generating said risks, but the social conditions themselves.

There is a discussion of socioeconomic vulnerability regarding the degree to which social classes are differently at risk. Recognizing the conditions of socioeconomic vulnerability as a generator of risk and disaster has a political dimension: it implies the acknowledgement that in the states of risks and disasters there is a political and social responsibility, which refers to the notions of environmental and spatial justice (*Campos-Vargas, Toscana-Aparicio & Campos-Alanís, 2014*).

It is important to emphasize that disasters are not only the products of threats, they are consequences of the environmentally unsustainable economic development model from which some sectors benefit, to the detriment of others, at all scales: global, national and regional (*Campos-Vargas, Toscana-Aparicio & Campos-Alanís, 2014*).

## Description of the area of study

### Location of the basin of the Santiaguillo Lagoon

The endorheic basin of Santiaguillo Lagoon is located to the north of the hydrological region no. 11 of the Presidio and San Pedro rivers, with a surface of 2,542.16 km$^2$. This basin partially covers the municipalities of Canatlán, Coneto de Comonfort, El Oro, San Juan del Río, Santiago Papasquiaro, and Nuevo Ideal (*Mexican National Commission of Water (CONAGUA), 2011*).

"Santiaguillo" is the southeast region from the route followed by the migratory birds that come from Alaska and Canada. These birds nest in Santiaguillo, which is why this site is considered one of the 30 most important wetlands. International organizations place the basin in an under-pressure water condition derived from the underground

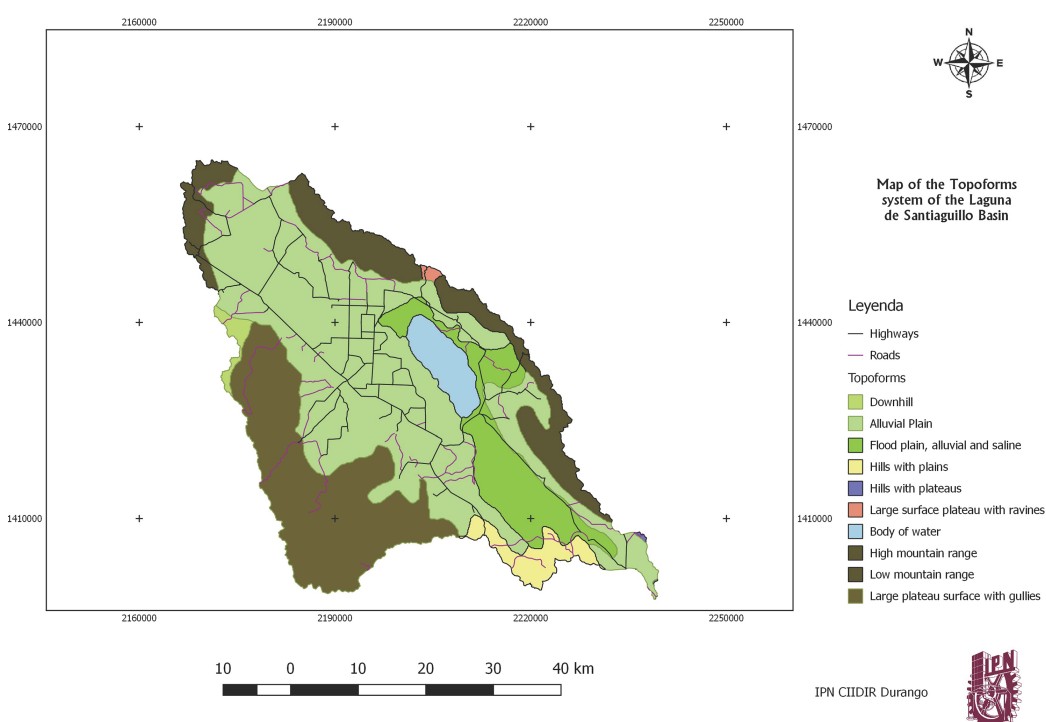

**Figure 1  Topoform system map.** Map using INEGI data.   

extraction, the pollution and the way to address the management of water and drought (*Mexican National Commission of Water (CONAGUA), 2012*).

### Topographic profile

Tracing a cut from south to north, the profile starts from a maximum altitude of almost 3,200 m.a.s.l. in the southeast zone, characterized by an accidental topography with steep slopes in the first 15 km. After that, there is an extension of 30 km where the slope is very small and finally there is an increase in altitude close to 2,000 m.a.s.l. in the north part.

### Orographic characteristics

The zone belongs to the physiographic province of the "Sierra Madre Occidental," and the sub-provinces are the Gran Meseta and the Cañones Duranguenses (Durango's ravines), which has its origin in the southern part to the west of the territory; the rest is described as mountains and plains of Durango (Fig. 1).

The basin is located inside a "graben," that is, a depression limited on both sides by raised faults, where the terrain of the internal part has sunk because of internal forces (*Nieto-Samaniego et al., 2012*), the area is characterized by the relief of the north and south zone that the surveys perfectly close the graben thus giving origin to the endorheic sub-basin (Fig. 2). Nieto describes the site as: "The Santiaguillo graben forms the northwestern part of the regional San Luis-Tepehuanes fault system, which is located between the Sierra Madre Occidental and the physiographic Mesa Central. In the region of Santiaguillo graben, we identified eight lithostratigraphic units with ages from Eocene to Quaternary" (*Nieto-Samaniego et al., 2012*).

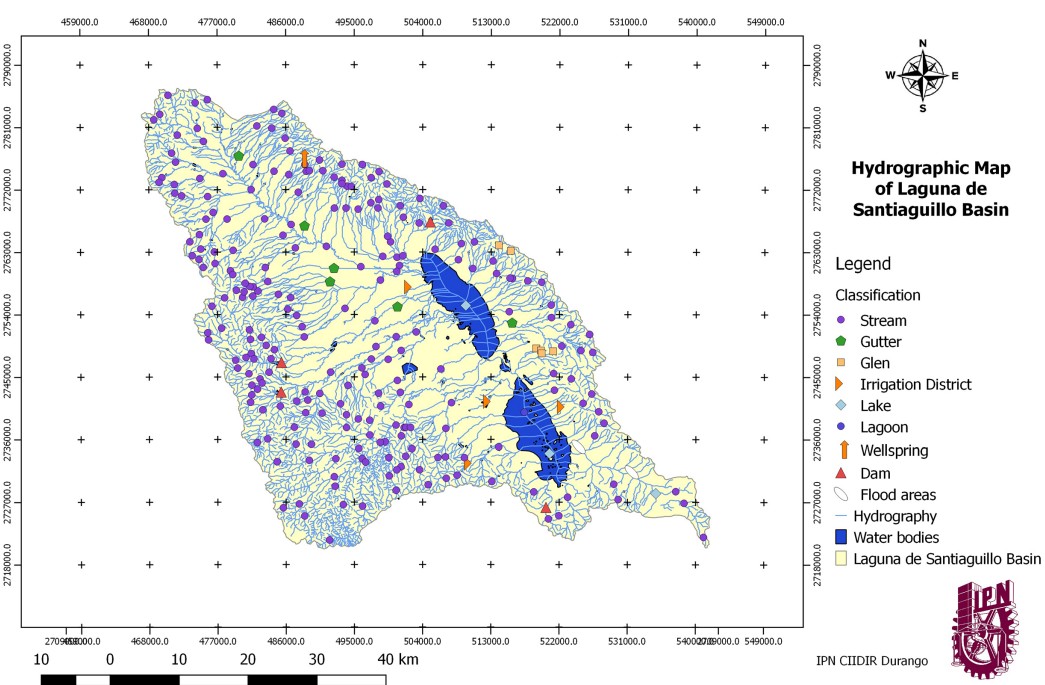

**Figure 2 Hydrograhic map of Laguna de Santiaguillo Basin.** Map created with vector information of INEGI (*Mexican National Institute of Statistics and Geography (INEGI), 2018*).

**Table 1 Types of climate and its extension in the sub-basin.**

| Climate | Hectares | Percentage (%) |
| --- | --- | --- |
| BS1kw(w) | 180,668.15 | 71.07 |
| C(E)(w1) | 14,206.50 | 5.59 |
| C(E)(w2) | 14,601.78 | 5.74 |
| C(w0) | 26,044.58 | 10.25 |
| C(w1) | 18,695.20 | 7.35 |

**Note:**
Table created with vector information from INEGI (National Institute of Statistics and Geography).

## Physical aspects

### Weather

In the valley of Santiaguillo, two types of climates are identified: dry semidesert (BS) and a series of mesothermal or temperate climates with dry period in winter (Cw). The temperature ranges from 8 to 18 °C, with an average precipitation of 300–800 mm (*Mexican National Institute of Statistics and Geography (INEGI), 2005*, *2010*).

Table 1 shows the extension of each type of climate, to get the classification of weather it was used the köppen classification.

### Hydrography

The map of Fig. 2 shows the detailed hydrography, which allows to understand the behavior shown in Fig. 3, where hydrology is presented by microbasins.

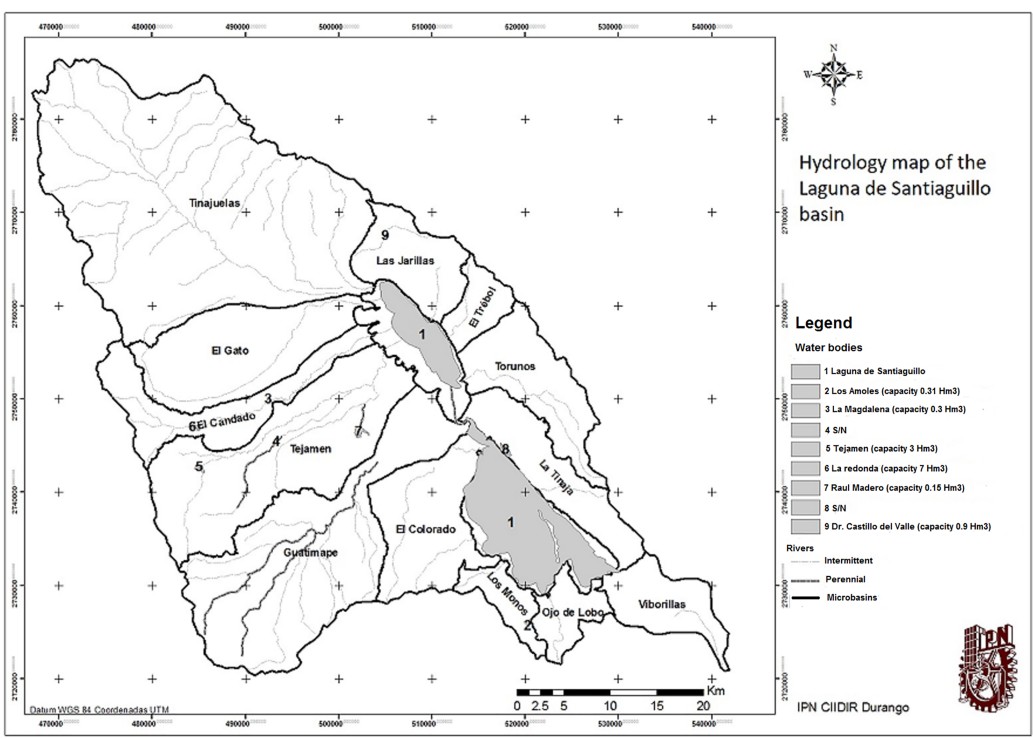

**Figure 3 Map of hydrology surface and division in micro-watersheds.** Map created with vector information of INEGI.

In this study case it is proposed the use of microbasins as a reference to the specific surface exploitation areas.

The most important micro-basin in the surface is located to the north, where the Tinajuelas stream is formed, this is the main runoff and it has its origin in the northern mountain range ending in the northern body of the Santiaguillo lagoon.

The streams of San Antonio, Grande, Hondo, San Francisco and the Tinajitas are its tributaries from the northeast. From the northeast area are also the streams: Los Alamos, Palos Colorados, Colorado and Salto de San Lucas. In this micro-basin it is also located the Zeta stream that rises in the northeast area and empties directly into the north lagoon; the stream of El Toboso has its origin in this area and it is the main contributor of water to the Zeta stream, which in turn joins the Tinajuelas at the mouth.

The Gato stream has its origin in Santa Rosa hill which passes through the side of the town of Nuevo Ideal and flows into the north lagoon. The Candado stream and Guatimape river or the Molino are born in the mountain range of Epazote.

Its runoff ends in two bodies of water located in the central part of the basin aligned from north to south, the topographic configuration around them is composed of areas of reduced extension, high and steep, the largest are located toward the southwest and northeast of the sub-basin; the topography of this site corresponds to mountain ranges of El Epazote, San Francisco and San Miguel, the altitude ranges fluctuates around 3,000 m.a.s.l., the average elevation is 2,160 m.a.s.l. forming a valley that has allowed the development of important productive activities (*Mexican National Commission of Water (CONAGUA), 2011*)

The streams in order of importance are Tinajuelas, Guatimape, Tejamen, El Gato, El candado, Torunos, Las Jarillas, and El Trebol; these streams feed the northern lagoon body that has perennial water. On the other hand, El Colorado, La Tinaja, Viborillas, Ojo de Lobo, and Los monos feed the southern lagoon which has an intermittent behavior.

The streams Alisos, Gigantes, and Seco lead into the Astilleros stream, which is another principal slope of the area. In the east area, there stand out the streams Hondo, Grande, Las Jarillas, and Coyotillos.

All the runoff forms the municipality of Nuevo Ideal and spills its waters into the northern lagoon.

In the municipality of Canatlán, the streams of La Soledad, Los Ladrones, Claveras, and La Tinaja have its slopes in the south lagoon.

### Surface lithology

The Laguna de Santiaguillo basin is made up of alluvial soils, which are in the plain areas, and cover 40.95%. These soils are known as alluvial fans, they are located next to mountainous belts and they are materials transported by water; their size varies from clay to thick gravels, ridges and blocks (*Servicio Geológico Mexicano, 1998*).

Rhyolite-acid tuff rocks cover 40.45% of the area and are characterized by volcanic rocks composed of quartz and alkali feldspar. A total of 11% of the surface rock is constituted by lacustrine sedimentary media which allows the presence of detrital and biodetritic, chemical, biochemical, and organic sediments; these are located mainly in the bed of both lagoon bodies, determining their important age (*Córdoba-Méndez, 1988*).

The rest of the geological composition of the area is Tufa-acid 3.13%, andesite 2.5% (volcanic rock), conglomerate 1.37% (sedimentary rock), wind 0.25% (the accumulation of materials has its origin in the wind), basalt 0.14% (volcanic igneous rock), intermediate tuff 0.06%, and tonalite 0.05% (plutonic igneous rock), the sub-basin distribution presents the most representative values.

### Soils

The soils that are distributed in the basin are of Vertisol type with 25.83% of the total area and it refers to a type of clay soil, heavy and fine texture, it is impermeable, susceptible to floods. It is hard to dry and tends to be a floor that turns or flips. Their presence reveals the lacustrine formation processes in the basin from the erosions of the middle and upper basin areas, depositing by dragging in the parts where the runoff speed is reduced due to the slope. It is present in temperate and warm climate, the vegetation on this type of soil in the basin consists of pastures and bushes.

Lithosol is present in 23.9% of the sub-basin and due to its acid characteristics and its thickness of about 10 cm of rock or limestone (tepetate), it is mainly used for grazing since it is not suitable for crops. The regosol type covers 19.33% of the sub-basin present on the west slope under the pine forests, it does not have horizons and it is a very permeable soil.

Solonchak is a type of salic soil characterized by being very permeable and it covers 8% of the territory of the sub-basin it is particularly identifiable in the unflooded lagoon beds. According to the vector data of the INEGI the water bodies do not apply in the matter of soil;

**Table 2 Soil surface by texture of the sub-basin.**

| Texture | Hectares | Percentage (%) |
|---|---|---|
| Medium | 141,793.72 | 55.78 |
| Fine | 77,436.36 | 30.46 |
| Gross | 22,909.04 | 9.01 |
| N/A water body | 12,077.11 | 4.75 |

Note:
Table created with vector data from INEGI.

their occupancy is 4.75%; however, when the south lagoon dries, it reveals this type of soil on its bed (*Mexican National Institute of Statistics and Geography (INEGI), 2004*; *da Silva, 1981*).

Regarding the texture of the soil, the sub-basin presents an average texture of 56% of its surface, followed by a fine texture with 30.46%; the coarse texture occupies 9.01% of the territory; the remaining 4.75% belongs to the southern lagoon and there is no data available regarding its texture (*Mexican National Institute of Statistics and Geography (INEGI), 2010*) (Table 2).

### Ground cover

In the sub-basin of Santiaguillo, there is vegetation of pine forests, most of it is located in the area of the plateau occupying 20.75% of the area, the forest of oak-pine occupies 3.5% and its distribution is in the eastern area, 2.43% oak forest is found in certain areas to the north, southeast and south of the sub-basin. The pine-oak forest occupies 1.32% and extends from the north to the east of the mountain range of Coneto (*Mexican National Institute of Statistics and Geography (INEGI), 2010*).

The area studied has three types of grassland: 10.45% of the surface of the Santiaguillo valley presents natural grassland, 3.47% induced pastureland and 2.58% is halophilic grassland. A total of 3.94% of the surface is crasicaule scrub, the bushes of fleshy stems such as nopales, Opuntia cactaceae, and others are dominant (*Mexican National Institute of Statistics and Geography (INEGI), 2010*).

### Description of the aquifer

The "Valle de Santiaguillo" aquifer has an average annual recharge of 50.7 million m$^3$/year, against a natural discharge of 8.0 million m$^3$/year. The concessioned volume of groundwater is 60 million m$^3$, and according to the CONAGUA it is in a deficit of $-15.472461$ million m$^3$/year (*Mexican National Commission of Water (CONAGUA), 2009*).

With regard to the administrative situation, it is active for exploitation and use of national waters and the aquifer in its entirety is free of delivery (*Mexican National Commission of Water (CONAGUA), 2009*).

According to the decree published on December 31, 1999, in the "Diario Oficial de la Federación," three zones are established for the availability of the aquifer. The 1,039 belonging to the municipality of Nuevo Ideal and located in the area of availability 6, 1,001 of the municipality of Canatlán within zone 6 of availability, and 1,032 of Santiago Papasquiaro within the area of availability 7.

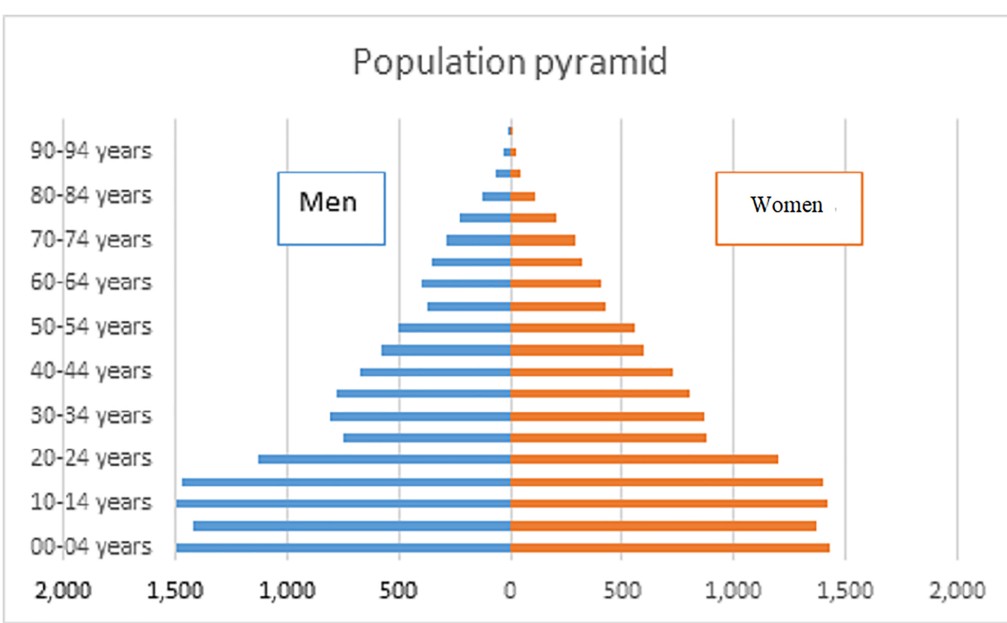

**Figure 4 Population pyramid of the Laguna de Santiaguillo basin.** Figure created using 2010 census information of INEGI.

CONAGUA establishes that the aquifer is in a state of vulnerability with an index value of 7, which in the scale refers to a high contamination vulnerability (*Mexican National Commission of Water (CONAGUA), 2009*).

## Economic and social aspects
### Demography

According to the results of the INEGI census and counting the 151 locations located within the basin, in a period of 10 years (2000–2010), 102 locations of the 131 had a decrease in their population; 41 of them showed an increase and nine maintained an equal number of inhabitants (*Mexican National Institute of Statistics and Geography (INEGI), 2010*).

Of the 31,967 inhabitants of the basin, 21,091 (66%) live in rural areas (towns with less than 2,500 inhabitants) and the rest live in the town of Nuevo Ideal with a population of 10,876 inhabitants (34%) (*Mexican National Institute of Statistics and Geography (INEGI), 2010*).

The analysis of the population pyramid reveals that the birth rate is higher in males than in females. To the 2,049 children registered under 2 years old, 965 were female and 1,084 were males, the difference remains constant until the age of the majority is exceeded since the range of 18-to-24-year-old women (2,076) exceeded the number of men (2,026) due to immigration seeking better employment opportunities. The third age (60 years and older) is 12% (3,872 individuals) of the total population (*Mexican National Institute of Statistics and Geography (INEGI), 2010*) (Fig. 4).

### Access to education

Regarding the literacy of the 24,123 people who were examined in this area, 1,952 persons (8%) do not attend school, in this category small children who are not yet of school age are

omitted, including children over 5 years and adolescents up to 14 years old. A total of 1,621 persons (7%) of the individuals whose ages vary between 15 and 24 years attend school, while 4% (919 people including children and adolescents ranging from 8 to 15 years old, and adults) were considered illiterate. Nevertheless, the figures indicate that almost 70% of the population has unfinished education, and in many cases incomplete basic education (*Mexican National Institute of Statistics and Geography (INEGI), 2010*).

### Economic conditions

Of the 13 largest locations in the basin (20,363 inhabitants), only 35% (7,062 inhabitants) are economically active. The first income for the families of these communities comes from men since 80% of the total economically active population is male (*Mexican National Institute of Statistics and Geography (INEGI), 2010*).

A total of 58% of the total population (18,593 inhabitants) are entitled to medical care and health services in some public or private institution, 41% of them do not to have the right to receive any medical attention, while a minority of 1% did not provide this information (*Mexican National Institute of Statistics and Geography (INEGI), 2010*).

### Access to public health services

Of the results obtained by the sample of the 13 locations with significant populations, 66% have medical services and 34% (20,363 inhabitants) do not (*Mexican National Institute of Statistics and Geography (INEGI), 2010*).

The total number of households in the 151 locations is 7,879, of which 6,122 are headed by a male or head of a family (78%) and 1,757 are female-headed (22%) (*Mexican National Institute of Statistics and Geography (INEGI), 2010*).

The households in which the head of family is male account for 81% of the total population, which corresponds to 25,690 inhabitants; the households whose head of family was a woman are 19% (5,959 inhabitants) (*Mexican National Institute of Statistics and Geography (INEGI), 2010*).

### Housing conditions

Of the 11,649 homes surveyed, 11,545 homes are private and only 7,959 (68%) are inhabited dwellings; the density is 4 inhabitants per house (*Mexican National Institute of Statistics and Geography (INEGI), 2010*).

Of the 7,879 inhabited private homes, 96% have a different floor than the land and only 4% have a dirt floor; 70% have two or more bedrooms; 90% (7,080) of the houses are made up of three or more rooms (kitchen, bathroom, living room, bedroom, etc.) (*Mexican National Institute of Statistics and Geography (INEGI), 2010*).

A total of 98% of the houses have electricity service, 94% have piped water service, 91% have a toilet, and 84% have drainage service connected to a municipal network, 80.3% of the total houses studied in this analysis have all of the aforementioned services (*Mexican National Institute of Statistics and Geography (INEGI), 2010*).

### Marginalization index

According to the data provided by CONAPO (Comisión Nacional de Población) in 2015, the municipality of Canatlán showed an index of marginalization of −0.822, Nuevo

Ideal −1.001 y Santiago Papasquiaro −0.676, which places the basin of the Laguna de Santiaguillo in a low degree of marginalization (*National Population Commission of Mexico (CONAPO), 2015*).

The aim of this article is to evaluate the availability of water in Laguna de Santiaguillo basin derived from the overexploitation of the aquifer to propose the main ideas to get some sustainable mitigation measures.

## MATERIALS AND METHODS

To get the characterization of the sub-basin, the data employed was vector and demographic obtained from the National Institute of Geography, Informatics and Statistics (INEGI) and it was processed using geographic information systems. The information of the hydraulic infrastructure of the principal rivers in the zone, the public register of the rights of water and some of the vector data were obtained from the National Commission of Water (CONAGUA). Meteorological information was collected from the register of National Meteorology Service (SMN) (https://smn.conagua.gob.mx/es/informacion-climatologica-por-estado?estado=dgo) and the spatial analysis was made using the computer package Arcgis 9.3.

In the field, the outline of the sub-basin in the plane zones was studied by giving a terrestrial travel verifying the watersheds corresponding to the limits of the sub-basin. A GPS Garming Oreon 550 was used, along with a UTM coordinate system with a Datum WGS84.

Each micro-basin was overlaid by layers putting together the natural data that influences the water availability. The vector layers of hydraulic infrastructure, roads, locations, and Agricola zone were used to get the anthropic factors and the influence they have in the zone following water availability. This allowed to attain knowledge of the natural dynamic and social conditions about the water use and cycle in the zone.

A hydrological balance was carried out at micro-basin level using the methodology indicated by the NOM-011- CONAGUA-2015 and the manual given by the Mexican Institute of Water Technology (IMTA).

The Nom 011 established the average annual availability of national waters in hydrological basins and in aquifers. The method shall be considered as the minimum mandatory technical requirement and does not exclude the additional application of complementary methods or the more complicated and precise cases, when the available information allows it, in which case the Commission shall review together with the users and determine which are the results that prevail.

The Norm basically signalize the availability annual average of surface water in the basin, annual average volume of wring of the basin toward waters down, availability annual average of water from the subsoil in an aquifer.

To determine the main channel at the outlet of the hydrological basin, by means of the following expression:

$$
\begin{aligned}
\text{Availability annual average of surface water in the basin} \\
= \text{Annual average volume of the basin runoff towards water down} \\
- \text{Current annual volume committed water down}
\end{aligned}
$$

**Table 3  Levels of water availability.**

| Availability of water in thousands m³/hab/year | Water availability levels |
|---|---|
| <1 | Catastrophically low |
| 1.1–2.0 | Very low |
| 2.1–5.0 | Low |
| 5.1–10 | Medium |
| 10.1–20 | High |
| >20 | Very high |

Note:
*Shiklomanov & Rodda (2004)*.

The mean annual runoff volume of the basin downstream of the site of interest is determined when applying the following expression:

Annual average volume of runoff of the basin towards waters down
= Annual average volume of runoff from the upper waters basin
+ Medium volume annual runoff natural + Annual volume of returns
+ Annual volume of imports − Annual volume of exports
− Annual volume of extraction of superficial water
− Annual average volume evaporation in reservoirs
− Medium volume annual variation storage in embalses

To determine the average annual availability of subsoil water in an aquifer apply the equation:

Availability annual average of water from the subsoil in an aquifer
= Total recharge annual average − discharge natural engaged
− water extraction groundwater

Combined with the regulations, the runoff coefficients given in the manual were used: evaluation of water resources, elaboration of water balance integrated by river basins.

Derived from the runoff coefficients, the volumes that infiltrate, evaporate and drain are evaluated.

Starting from the scenario given by the balance of surface and groundwater other scenarios were modeled by simulating the reduction of extraction to find the ideal conditions.

The analysis was based on the concept of natural runoff. Based on the above, the relative availability understood as the ratio between availability and total human and natural demand was determined. Under the usual criteria, a relative availability under 1.4 is classified as a deficit, from 1.4 to 3.0 is qualified as equilibrium, from 3.0 to 9.0 as availability and greater than 9 as abundance.

In this study case it is proposed the use of microbasins as a reference to the specific surface exploitation areas, they could be linked in an specific area in the aquifer. Even as it

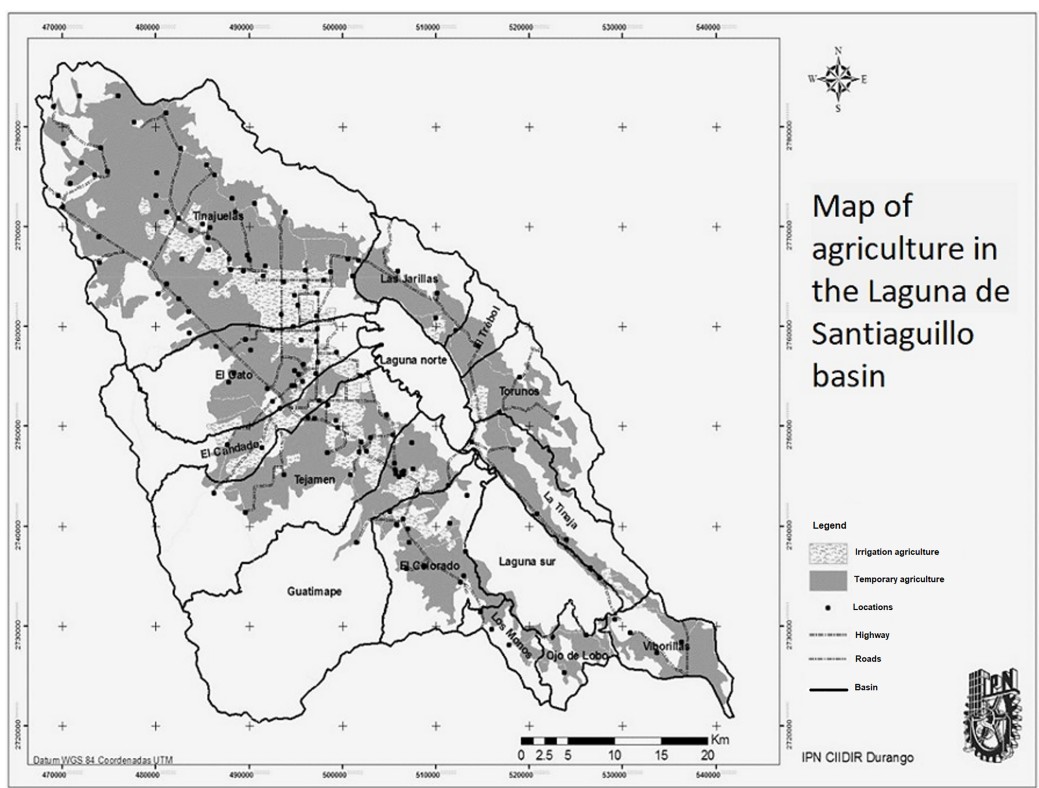

**Figure 5 Land use for agriculture in the sub-basin.** Map created with vector information of INEGI.

**Table 4 Number and volume of the underground sources granted.**

| Use | Number of wells | Concession volume thousands of m$^3$ |
|---|---|---|
| Agricultural | 1,329 | 5,233.70 |
| Livestock | 464 | 360.36 |
| Domestic | 49 | 20.76 |
| Urban public | 91 | 1,959.75 |
| Multiple | 26 | 607.86 |
| Different uses | 2 | 19.88 |
| Industrial | 5 | 6.87 |

**Note:**
Table created with information from the Public Register of Water Rights, of the National Commission of Water.

is just one aquifer, there are some areas in which the pressure is higher than in other areas, so it is propose the needed to regionalizer with the use of the microbasins.

To delimit the microwatersheds their areas were determined, and the corresponding cuttings were made in all the vector layers to generate the detailed cartography.

From the topographic mapping analysis, it was determined that the lake surface reaches 8% of the total basin.

The water availability criteria used are those presented by Shiklomanov (Table 3).

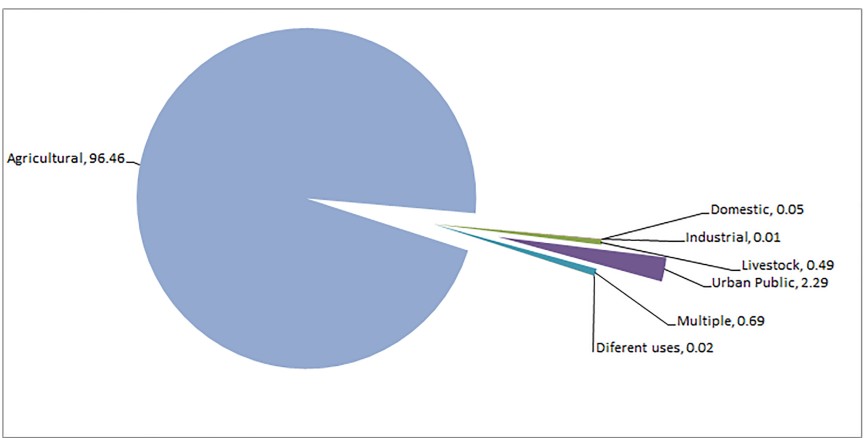

**Figure 6 Use of groundwater in the Laguna de Santiaguillo basin (%).** Figure created with CONAGUA information.

## RESULTS

### Land use

According to INEGI'S vector data 2012, the sub-basin presents the following land use (Fig. 5): 51.08% of the area has a forest, livestock or agricultural use, this area covers the plains region of agricultural use; 17.54% is considered for irrigation farming is at the largest area found in the north. In the western zone, there is the Tejamen dam, where an extension of irrigated farmland originates and reaches the central area of the sub-basin, covering the needs either by running water or by the series of wells found in the zone according to the vector data of the REPDA, in the same way the dam of El Candado presents a portion of irrigated farmland by water rolled in combination with wells of water in the area, 82.46% is seasonal agriculture (Table 4).

To obtain the map of Fig. 6 the information was used, the agricultural information provided by INEGI was used, the information used corresponds to the year 2012.

### Hydrological balance of surface water

The determination of the relative availability at the micro-watershed level reflects a regionalized manner of the different pressure conditions to which the resource is subject due to human activity.

All the determinations were based on the volumes concessioned and registered in the Public Registry of Water Rights of the National Water Commission; therefore, the balance reflects only the administrative condition and not necessarily the real one: however, in the absence of data regarding the real extractions of the surface water volumes used and those extracted from groundwater. This study allows us to evaluate the condition of vulnerability of the basin based on the availability of the resource due to the way in which use is regulated.

In the micro-watersheds that contribute their runoff to the northern part of the Santiaguillo lagoon, relative availability is obtained with deficit conditions in the micro-basins: "Presa Tejamen" to the east of the lagoon body and "Torunos" located to the west, following the other micro-basins exert a little pressure over the water resource.

**Table 5 Results of the balance surface waters of the micro-basins of the Laguna de Santiaguillo.**

| Name of the hydrological sub-basin | Cp.- annual average volume of natural runoff | Ab.- average annual volume of runoff from the basin downstream | Rxy.- current annual committed downstream | Ab-Rxy | D.- average annual availability of surface water in the hydrological basin | Classification of relative availability |
|---|---|---|---|---|---|---|
| Guatimape | 18.66 | 18.24 | 10.72 | 7.52 | 7.52 | Balance |
| P Tejamen | 3.75 | 3.61 | 7.24 | −3.63 | −4.51 | Deficit |
| Tejamen | 17.46 | 12.58 | 7.24 | 5.34 | 8.95 | Balance |
| El Gato | 14.81 | 14.72 | 5.57 | 9.15 | 9.15 | Balance |
| Tinajuelas | 37.35 | 36.81 | 21.20 | 15.61 | 15.61 | Balance |
| Presa Dr. Castillo del Valle | 0.50 | 0.42 | 0.02 | 0.40 | 0.31 | Abundance |
| Las Jarillas | 4.53 | 4.52 | 2.32 | 2.20 | 2.61 | Balance |
| El Trébol | 2.01 | 2.00 | 1.23 | 0.76 | 0.76 | Balance |
| Torunos | 5.15 | 5.08 | 3.67 | 1.42 | 1.42 | Deficit |
| Presa La Redonda | 3.54 | 3.33 | 0.17 | 3.16 | 2.95 | Abundance |
| El Candado | 9.72 | 9.54 | 3.65 | 5.90 | 9.23 | Availability |
| Laguna Norte | 44.18 | 11.36 | – | 11.36 | 102.13 | – |
| Ojo de Lobo | 2.68 | 2.66 | 2.67 | −0.01 | −0.01 | Deficit |
| Tinaja | 5.26 | 4.87 | 6.32 | −1.45 | −1.45 | Deficit |
| Los Monos | 2.12 | 2.06 | 2.24 | −0.18 | −0.18 | Deficit |
| El colorado | 6.79 | 6.42 | 8.49 | −2.07 | −2.07 | Deficit |
| Viborillas | 4.86 | 4.83 | 4.47 | 0.36 | 0.36 | Deficit |
| Laguna Sur | 72.64 | 48.73 | – | 48.73 | 93.47 | – |

**Note:**
Results of the balance elaboration using the method given by the *Mexican Diario Oficial de la Federación (2015)*.

Abundance conditions are only found in the micro-basin located upstream of the Dr. Castillo del Valle and La Redonda dams, which due to their natural conditions are important generators of water that are not used. Moreover, in the case of La Redonda there are conservation actions in the upper basin.

In the micro-basins that provide water to the southern part of the Laguna de Santiaguillo with significantly smaller extensions, with human presence in practically all of them and with a high pressure on a very limited resource, the general conditions of relative availability are deficit.

The results of the surface water balance (Table 5), in combination with the groundwater balance (Table 6), describe the heterogeneity of the basin that favors differences in the amount of rainfall within it. According to the calculations made, an annual total precipitated volume of 1,416 million m$^3$ is estimated.

The texture of the soil and the combination with the differences in the vegetation cover of the zone propitiate a variation in the drained volume, the coefficient of average draining that was determined is 10.45%.

The average annual volume of runoff was estimated at 341 million m$^3$. The Gato, Tejamen, and El Candado micro-basins present higher runoff coefficients due to the areas they use for agricultural activity.

**Table 6 Results of the analysis of groundwater availability by micro-basin in the aquifer in Laguna de Santiaguillo.**

| Micro-basin | Rain infiltration | Infiltration for returns | Horizontal entry | Evapo-transpiration | Total annual average recharge | Natural discharge committed | Subscribed underground volume | Underground availability | Exploitation/recharge (%) |
|---|---|---|---|---|---|---|---|---|---|
| | Ip | Ir | Eh | Evt | | | B | Dm | |
| Guatimape | 0.84 | 0.32 | 2.10 | 182.67 | 3.25 | 182.67 | 4.55 | −183.97 | 140 |
| Tejamen | 1.27 | 0.90 | 1.01 | 115.46 | 3.17 | 115.46 | 12.38 | −124.67 | 390 |
| El Gato | 1.09 | – | 0.41 | 98.52 | 1.49 | 98.52 | 11.68 | −108.71 | 783 |
| Tinajuelas | 5.37 | – | 2.04 | 314.06 | 7.41 | 314.06 | 40.75 | −347.41 | 550 |
| Las Jarillas | 0.77 | – | 0.31 | 36.01 | 1.08 | 36.01 | 6.57 | −41.50 | 608 |
| El Trébol | 0.50 | – | 0.15 | 19.88 | 0.65 | 19.88 | 0.75 | −19.98 | 116 |
| Torunos | 1.09 | – | 0.41 | 49.90 | 1.51 | 49.90 | 5.55 | −53.94 | 369 |
| El Candado | 0.40 | – | 0.48 | 63.77 | 0.88 | 63.77 | 4.23 | −67.12 | 478 |
| Ojo de Lobo | 0.26 | – | 0.18 | 0.24 | 0.44 | 0.24 | 0.00 | 0.20 | 0 |
| Tinaja | 0.56 | – | 0.30 | 0.52 | 0.86 | 0.52 | 1.64 | −1.30 | 191 |
| Los Monos | 0.21 | – | 0.18 | 0.19 | 0.39 | 0.19 | 0.02 | 0.18 | 6 |
| El Colorado | 1.21 | – | 0.43 | 1.12 | 1.64 | 1.12 | 6.75 | −6.23 | 412 |
| Viborillas | 0.32 | – | 0.14 | 0.30 | 0.46 | 0.30 | 0.03 | 0.14 | 6 |

**Note:**
Analysis of groundwater availability by micro-basin.

The surface lithology in combination with the types of soil and its texture allow an average infiltration percentage of 1.84% of the total precipitation, the total annual average volume infiltrated to the aquifer was calculated in 29.18 million $m^3$. The infiltration is attributed to the types of soil and rock present in the micro-basin, it was observed that areas with less infiltration have a combination of soils of the vertisol or solonchak type and tuffs.

## Volumes of concessioned water

The area studied presented different volumes of surface water concessions, highlighting the El Candado, Tejamen, and Las Jarillas micro-basins with the most important hydraulic works and surface concessions in the area.

Based on the results of the integral hydrological balance, it was observed that the micro-basins are capable of capturing sufficient rainfall to satisfy the surface granted by CONAGUA; however, two of them showed a deficit since their contribution is not enough to provide the volumes awarded. Specifically in relation to the El Candado micro-basin, the results obtained showed surface water availability conditions.

The average runoff in the basin is 135.79 millions of $m^3$/year, the shallowness of the superficial and underground natural reservoirs in the sub-basin favors the evaporation of 81 millions of $m^3$/year. Added to this the geographical conditions of the sub-basin are short-term streams and a quick discharge to the body of water called the Santiaguillo lagoon, only 16.26 millions of $m^3$ have been concessioned in the middle and upper basin. With this, the surface availability is 38.53 millions of $m^3$, thus resulting in a relative availability of 1.4 considered as deficit (*Mexican National Commission of Water (CONAGUA), 2017a*).

It was estimated that the natural surface availability per capita (amount of water that exists) can be allocated to their economic activities, which was obtained by dividing the drained volume between the number of inhabitants. Most micro-basins showed levels ranging from average to high. Based on this data per person, the quantity of water used in different activities is obtained over 6,000 m$^3$ on an annual basis per inhabitant.

The availability of development was determined by calculating the difference between the drained volume with the surface water concessioned, to know if they have the capacity to provide surface water to cover future demands in economic activities such as livestock or agriculture. The micro-basins that provided sufficient surface water to satisfy demands in terms of agriculture and livestock are El Candado, Guatimapé, Viborillas, and Ojo de Lobo whose values of excess drained water of the volume concessioned propitiate this condition.

## Hydraulic Infrastructure

In the area, there are three small storage dams for agricultural irrigation use: La Redonda, Tejamen, and Dr. Castillo del Valle. They have storage capacities of 7.0, 3.0, and 0.8 Hm$^3$.

There is a significant number of deep wells that according to the public registry of water rights (REPDA) amount to 1,958 works in the cut to 2011 (*Mexican National Commission of Water (CONAGUA), 2011*).

## Hydrological balance of groundwater

### Volume of concessioned groundwater

The aforementioned conditions also influence that the Santiaguillo Valley aquifer receives an average annual recharge of 30.42 millions of m$^3$/year with a committed natural discharge of 23.04 millions of m$^3$/year, mostly by evapotranspiration. The aquifer presents concessions of 90.69 millions of m$^3$, almost three times the capacity of the average annual recharge, which favors overexploitation, so that the underground deficit is 83.32 millions of m$^3$/year (*Mexican National Commission of Water (CONAGUA), 2002*, *2011*).

The number of underground extraction points counted for this study is 1,996 wells, distributed throughout the lower part of the basin around the lagoon bodies. The concessions were concentrated in Table 4, converting agricultural use as the main groundwater use with 1,329 wells and over five million m$^3$ of annual concessions.

The use of groundwater in the Laguna de Santiaguillo basin has its greatest application in agriculture; the percentages of the use are showed in Fig. 6.

The analysis of the infiltration capacity of the micro-basins in comparison with the exploitation of the aquifer supposes extractions that exceed their recharge capacity. A higher level of exploitation was observed in the micro-basins located to the west and north, exceeding more than five times its recharge capacity.

Given the condition of overexploitation of the aquifer and its limited recharge conditions, the modeling of scenarios was carried out, which allowed to observe whether the expectations of resolving the vulnerability of the basin are reachable or not.

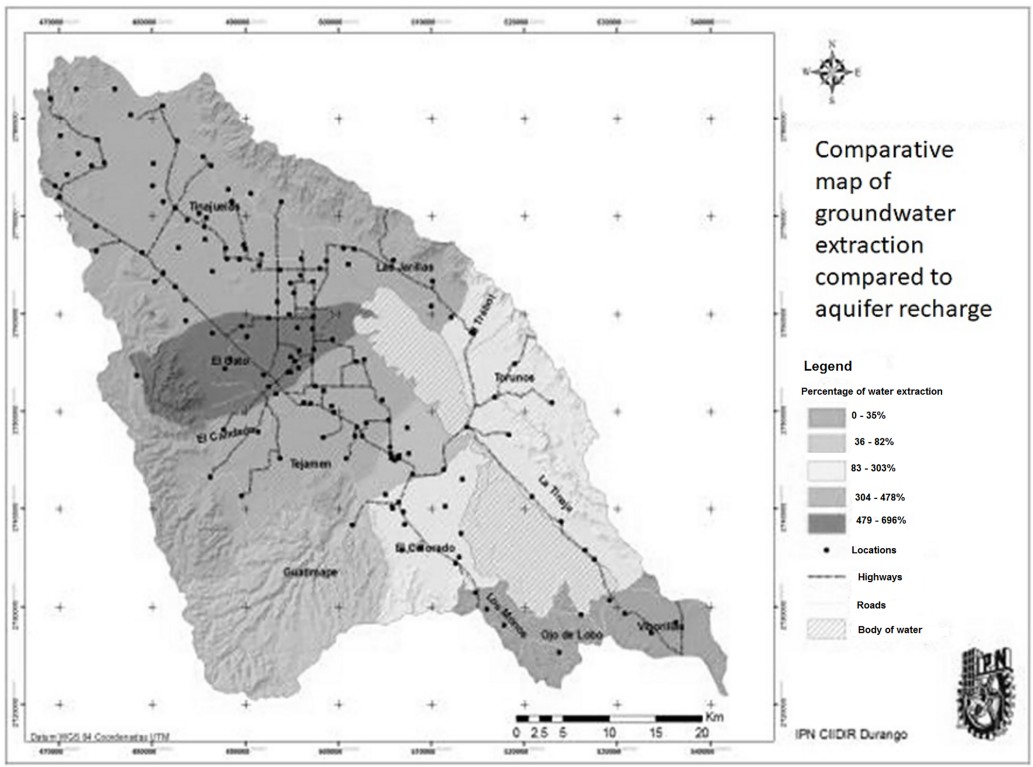

**Figure 7 Map of aquifer overexploitation by micro-basins.** Overexploitation by micro-basins.

To do this, the five scenarios were modeled making a reduction of extractions:

## Socio-environmental vulnerability assessment

### Delimitation of micro-basin

For the purposes of this investigation, the basin area was divided into 13 micro-basins of the mentioned streams, as well as the lagoon bodies (Fig. 7).

From the topographic mapping analysis, it was determined that the lake surface reaches 8% of the total basin and, according to this, the northern lagoon that covers only 81.38 Km$^2$ (3.2%) is fed with the runoff of 1,905.34 Km$^2$, that is to say, 75% of the surface of the sub-basin and the southern lagoon that covers 136.21 Km$^2$ (5.4%) feeds on the runoff of 16.5%, which in part explains the fact that the north lagoon has perennial water and the southern lagoon has a rather intermittent behavior.

## Analysis of scenarios

### Scenario zero

It arises from the hydrological balance made only with the natural conditions of rainfall and runoff without taking into consideration the infrastructure and giving zero value to human uses, this allows having an overview of the natural condition of the micro-basins in order to locate the criteria of analysis. The results obtained in the indicators for this scenario are presented in Table 7.

**Table 7  Availability indicators in response to the zero scenario.**

| Micro-basin | Natural surface availability per capita m³ | Level of availability per capita natural water (*Shiklomanov & Rodda, 2004*) | Availability for development m³ | Level of availability for development (*Shiklomanov & Rodda, 2004*) | Natural availability underground per capita m³ | Level of availability per capita natural water (*Shiklomanov & Rodda, 2004*) | Availability for development m³ | Level of availability per capita natural water (*Shiklomanov & Rodda, 2004*) |
|---|---|---|---|---|---|---|---|---|
| Guatimapé | 13,460.85 | High | 7,791.03 | Medium | 2,116.60 | Low | 1,768.71 | Very low |
| Tejamen | 6,075.61 | Medium | 4,172.20 | Low | 652.01 | Catastrophically low | 601.62 | Catastrophically low |
| El Gato | 1,210.52 | Very low | 873.74 | Catastrophically low | 121.97 | Catastrophically low | 65.67 | Catastrophically low |
| Tinajuelas | 4,021.41 | Low | 2,312.49 | Low | 797.38 | Catastrophically low | 737.78 | Catastrophically low |
| Las Jarillas | 3,014.57 | Low | 1,949.25 | Very low | 647.09 | Catastrophically low | 328.32 | Catastrophically low |
| El Trébol | 5,411.61 | Medium | 2,974.21 | Low | 1,736.09 | Very low | 1,204.08 | Very low |
| Torunos | 5,839.21 | Medium | 2,756.27 | Low | 1,707.23 | Very low | 160.19 | Catastrophically low |
| El Candado | 14,288.59 | High | 11,031.89 | High | 952.73 | Catastrophically low | 837.89 | Catastrophically low |
| Ojo de Lobo | 536,810.28 | Very high | 205,749.29 | Very high | 87,345.81 | Very high | −123,176.01 | Catastrophically low |
| La Tinaja | 6,063.64 | Medium | 1,169.80 | Very low | 992.68 | Catastrophically low | −745.11 | Catastrophically low |
| Los Monos | 41,663.46 | Very high | 13,562.84 | High | 7,659.71 | Medium | −3,163.80 | Catastrophically low |
| El Colorado | 3,852.35 | Low | 704.43 | Catastrophically low | 929.50 | Catastrophically low | 627.87 | Catastrophically low |
| Viborillas | 41,149.90 | Very high | 17,949.19 | High | 3,932.24 | Low | −3,096.35 | Catastrophically low |

**Note:**
Table created using Shiklomanov values.

### Status quo scenario

The status quo scenario (Table 8) corresponds to the integrated hydrological balance of the current conditions of the basin through the analysis of the water resource uses reported by the National Water Commission (CONAGUA) in the Public Registry of Water Rights.

From this analysis, it can be summarized that the basin currently has an groundwater use of 318% compared to the annual volume infiltrated to the aquifer; hence its condition of overexploitation, the micro-basins that currently have the greatest pressure on the aquifer are: Tejamen (419%), El Gato (721%), Tinajuelas (478%), Las Jarillas (356%), El Padrón (412%), and El Colorado (303%).

### Low scenario

The results of the low scenario are presented in Table 9, under the schemes of normative regulation of groundwater extraction for agricultural use mainly, low and medium

**Table 8  Availability indicators in response to the status quo scenario.**

| Micro-basin | Natural surface availability per capita m³ | Level of availability per capita natural water | Availability for development m³ | Level of availability for development | Natural availability underground per capita m³ | Level of availability per capita natural water | Availability for development m³ | Level of availability per capita natural water |
|---|---|---|---|---|---|---|---|---|
| Guatimapé | 13,460.85 | High | 7,428.39 | Medium | 2,237.48 | Low | 60.08 | Catastrophically low |
| Tejamen | 6,075.61 | Medium | 2,990.30 | Low | 826.17 | Catastrophically low | −2,684.85 | Catastrophically low |
| El Gato | 1,210.52 | Very low | 932.75 | Catastrophically low | 150.89 | Catastrophically low | −992.72 | Catastrophically low |
| Tinajuelas | 4,021.41 | Low | 2,597.11 | Low | 989.47 | Catastrophically low | −3,800.38 | Catastrophically low |
| Las Jarillas | 3,014.57 | Low | 1,748.41 | Very low | 785.17 | Catastrophically low | −2,331.70 | Catastrophically low |
| El Trébol | 5,411.61 | Medium | 3,021.53 | Low | 1,797.15 | Very low | −2,990.13 | Catastrophically low |
| Torunos | 5,839.21 | Medium | 2,776.44 | Low | 1,922.16 | Very low | −3,960.13 | Catastrophically low |
| El Candado | 14,288.59 | High | 6,778.18 | Medium | 1,115.25 | Very low | −3,597.86 | Catastrophically low |
| Ojo de Lobo | 536,810.28 | Very high | 205,340.18 | Very high | 87,345.81 | Very high | −123,362.21 | Catastrophically low |
| La Tinaja | 6,063.64 | Medium | 1,105.08 | Very low | 1,054.22 | Very low | −3,228.21 | Catastrophically low |
| Los Monos | 41,663.46 | Very high | 13,587.91 | High | 7,676.18 | Medium | −3,382.63 | Catastrophically low |
| El Colorado | 3,852.35 | Low | 871.35 | Catastrophically low | 1,048.01 | Very low | −2,432.57 | Catastrophically low |
| Viborillas | 41,149.90 | Very high | 17,802.71 | High | 3,932.24 | Low | −4,464.37 | Catastrophically low |

**Note:**
Table created using NOM-011-CONAGUA-2015 and IMTA manual.

technification in irrigation systems. Proposing a reduction order of 20% of the losses and waste in these activities, particularly those related to underground extraction.

The response of the basin in this scenario is an exploitation of the aquifer in 258% compared to the recharge capacity, where the greatest impact is produced in the micro-watersheds: Tejamen (349%), El Gato (699%), Tinjuelas (398%), Las Jarillas (295%), Padlock (337%), and Colorado (248%), where there is a considerable decrease in the exploitation of the aquifer.

### Medium scenario

It was carried out under a 65% extraction of the current annual volume concessioned. The response of the basin in terms of groundwater is 210% exploitation of the recharge capacity of the aquifer. Under this criterion, the exploitation by micro-basins is as follows: Tejamen has an underground exploitation of 292%, El Gato shows an extraction of 502%, Tinajuelas explodes in 333%, Las Jarillas presents an underground concession of

**Table 9 Availability indicators in response to the low scenario.**

| Micro-basin | Natural surface availability per capita m³ | Level of availability per capita natural water | Availability for m³ development | Level of availability for development | Natural availability underground per capita m³ | Level of availability per capita natural water | Availability for development m³ | Level of availability per capita natural water |
|---|---|---|---|---|---|---|---|---|
| Guatimapé | 13,460.85 | High | 7,396.70 | Medium | 2,214.03 | Low | 402.54 | Catastrophically low |
| Tejamen | 6,075.61 | Medium | 2,931.51 | Low | 794.29 | Catastrophically low | −2,024.61 | Catastrophically low |
| El Gato | 1,210.52 | Very low | 920.26 | Catastrophically low | 145.11 | Catastrophically low | −781.03 | Catastrophically low |
| Tinajuelas | 4,021.41 | Low | 2,538.73 | Low | 951.06 | Catastrophically low | −2,892.74 | Catastrophically low |
| Las Jarillas | 3,014.57 | Low | 1,706.35 | Very low | 758.23 | Catastrophically low | −1,799.02 | Catastrophically low |
| El Trébol | 5,411.61 | Medium | 3,004.28 | Low | 1,784.94 | Very low | −2,151.29 | Catastrophically low |
| Torunos | 5,839.21 | Medium | 2,719.68 | Low | 1,879.52 | Very low | −3,135.72 | Catastrophically low |
| El Candado | 14,288.59 | High | 6,734.41 | Medium | 1,091.64 | Very low | −2,701.81 | Catastrophically low |
| Ojo de Lobo | 536,810.28 | Very high | 205,340.18 | Very high | 87,345.81 | Very high | −123,362.21 | Catastrophically low |
| La Tinaja | 6,063.64 | Medium | 1,085.84 | Very low | 1,042.16 | Very low | −2,731.34 | Catastrophically low |
| Los Monos | 41,663.46 | Very high | 13,587.91 | High | 7,676.18 | Medium | −3,382.63 | Catastrophically low |
| El Colorado | 3,852.35 | Low | 836.78 | Catastrophically low | 1,024.31 | Very low | −1,820.48 | Catastrophically low |
| Viborillas | 41,149.90 | Very high | 17,802.71 | High | 3,932.24 | Low | −4,464.37 | Catastrophically low |

Note:
Table created using Shklomanov criteria.

246%, El Candado exploits 278%, and for the Colorado the groundwater resource is submitted to 205%, all percentages are in relation to the recharge capacity of the aquifer (Table 10).

### Ideal scenario

It departs from the integrated hydrological balance where the reduction of use is sought until achieving a balance between production and use of water. This scenario allows visualizing the conditions that would lead to sustainability in order to evaluate its viability, in which the extractions are equal to the infiltrations of the basin (Table 11).

### Viability of scenarios

Figure 8 shows the impact of the proposals of this investigation where it is observed that the viable scenario allows to reduce the pressure in the aquifer going from 318% of extraction to 258% if the technification of irrigation and good practices are

**Table 10 Availability indicators in response to the medium scenario.**

| Micro-basin | Natural surface availability per capita m³ | Level of availability per capita natural water | Availability for development m³ | Level of availability for development | Natural availability underground per capita m³ | Level of availability per capita natural water | Availability for development m³ | Level of availability per capital natural water |
|---|---|---|---|---|---|---|---|---|
| Guatimapé | 13,460.85 | High | 7,372.93 | Medium | 2,196.45 | Low | 659.38 | Catastrophically low |
| Tejamen | 6,075.61 | Medium | 2,887.43 | Low | 770.38 | Catastrophically low | −1,529.43 | Catastrophically low |
| El Gato | 1,210.52 | Very low | 910.89 | Catastrophically low | 140.78 | Catastrophically low | −622.27 | Catastrophically low |
| Tinajuelas | 4,021.41 | Low | 2,494.95 | Low | 922.25 | Catastrophically low | −2,212.02 | Catastrophically low |
| Las Jarillas | 3,014.57 | Low | 1,674.81 | Very low | 738.02 | Catastrophically low | −1,399.51 | Catastrophically low |
| El Trébol | 5,411.61 | Medium | 2,991.35 | Low | 1,775.78 | Very low | −1,522.15 | Catastrophically low |
| Torunos | 5,839.21 | Medium | 2,677.11 | Low | 1,847.54 | Very low | −2,517.41 | Catastrophically low |
| El Candado | 14,288.59 | High | 6,701.58 | Medium | 1,073.94 | Very low | −2,029.77 | Catastrophically low |
| Ojo de Lobo | 536,810.28 | Very high | 205,340.18 | Very high | 87,345.81 | Very high | −123,362.21 | Catastrophically low |
| La Tinaja | 6,063.64 | Medium | 1,071.41 | Very low | 1,033.12 | Very low | −2,358.68 | Catastrophically low |
| Los Monos | 41,663.46 | Very high | 13,587.91 | High | 7,676.18 | Medium | −3,382.63 | Catastrophically low |
| El Colorado | 3,852.35 | Low | 810.84 | Catastrophically low | 1,006.53 | Very low | −1,361.41 | Catastrophically low |
| Viborillas | 41,149.90 | Very high | 17,802.71 | High | 3,932.24 | Low | −4,464.37 | Catastrophically low |

**Note:**
Table elaborated using NOM-011-CONAGUA-2015 and Shiklomanov criteria.

adopted, reducing the current extraction to 20%, which is not economically and socially feasible.

***Effects on the potential social and economic development of the sub-basin.***
Analyzing the human activities that take advantage of the water in the basin could be reflected in the balances of surface and groundwater. It can be deduced that the micro-basins of the north area have small volumes of surface availability that are not very significant. The micro-basins of the south zone due to their small surface area, the dominance of temperate semi-dry climate and the strong relation of the area used for agriculture present deficits in all of them.

In contrast, the revision of the volumes of groundwater reflects an overexploitation of the aquifer in relation to recharge ranging from 116% to 738% in 10 of the 13 micro-basins.

In general, the high evaporation in the basin, a low average rainfall, and consequently a limited recharge of the aquifer also reduced by the high presence of alluvial soils in the

**Table 11  Availability indicators in response to the ideal scenario.**

| Micro-basin | Natural surface availability per capita m³ | Level of availability per capita natural water | Availability for development m³ | Level of availability for development | Natural availability underground per capita m³ | Level of availability per capita natural water | Availability for development m³ | Level of availability per capital natural water |
|---|---|---|---|---|---|---|---|---|
| Guatimapé | 13,460.85 | High | 7,317.48 | Medium | 2,155.41 | Low | 1,258.67 | Very low |
| Tejamen | 6,075.61 | Medium | 2,784.56 | Low | 714.58 | Catastrophically low | −374.00 | Catastrophically low |
| El Gato | 1,210.52 | Very low | 889.04 | Catastrophically low | 130.67 | Catastrophically low | −251.82 | Catastrophically low |
| Tinajuelas | 4,021.41 | Low | 2,392.80 | Low | 855.02 | Catastrophically low | −623.66 | Catastrophically low |
| Las Jarillas | 3,014.57 | Low | 1,601.20 | Very low | 690.88 | Catastrophically low | −467.32 | Catastrophically low |
| El Trébol | 5,411.61 | Medium | 2,961.17 | Low | 1,754.41 | Very low | −54.18 | Catastrophically low |
| Torunos | 5,839.21 | Medium | 2,577.78 | Low | 1,772.92 | Very low | −1,074.69 | Catastrophically low |
| El Candado | 14,288.59 | High | 6,624.99 | Medium | 1,032.63 | Very low | −461.69 | Catastrophically low |
| Ojo de Lobo | 536,810.28 | Very high | 205,340.18 | Very high | 87,345.81 | Very high | −123,362.21 | Catastrophically low |
| La Tinaja | 6,063.64 | Medium | 1,037.73 | Very low | 1,012.03 | Very low | −1,489.16 | Catastrophically low |
| Los Monos | 41,663.46 | Very high | 13,587.91 | High | 7,676.18 | Medium | −3,382.63 | Catastrophically low |
| El Colorado | 3,852.35 | Low | 750.34 | Catastrophically low | 965.05 | Catastrophically low | −290.26 | Catastrophically low |
| Viborillas | 41,149.90 | Very high | 17,802.71 | High | 3,932.24 | Low | −4,464.37 | Catastrophically low |

**Note:**
Table created using NOM-011-CONAGUA-2015 and Shiklomanov criteria.

lower part of the basin. Added to the conditions of the endorheic basin for not being connected to the rest of the territory, it generates an isolated space characterized by a fragile condition for the sustainability of human activities, considering the great agricultural and livestock vocation of the region where these activities are also based on the coexistence of cultures Mennonite and Mexican with deep-rooted traditions on how to take advantage of intensive natural resources.

This combination of cultural practices and limited availability both on the surface and in the aquifer could be understood as an intense pressure on the water resource, this generates a condition of fragility to the sustainability of production systems based on water consumption (Fig. 8).

## DISCUSSION

The high pressures on water resources have caused a great number of problems of a political, economic, social, and environmental nature (*Senet-Aparicio, Pérez-Sánchez & Bielsa-Artero,*

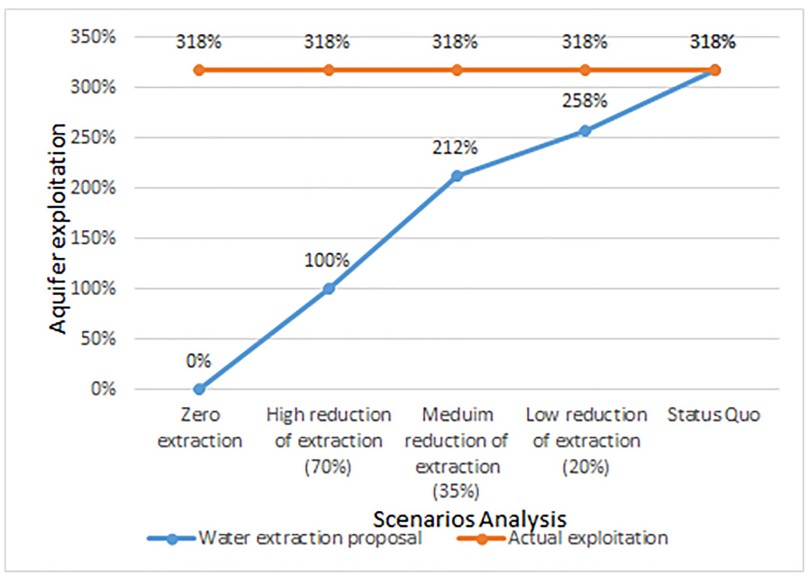

**Figure 8 Comparison of extraction reduction scenarios against overexploitation of the aquifer.** Comparison of extraction reduction scenarios.

). Therefore, it is very important to carry out actions to ensure the sustainability of the aquifer and the society that lives in the Laguna de Santiaguillo watershed.

The physiographic and socioeconomic conditions of the region are extremely important to know the degree of vulnerability of the aquifer. There are political decisions that generate important economic spillovers, but at a high environmental cost; an example of this is what happened in the Disi aquifer in Jordan.

Toward the year of 1984, the Jordanian government allowed the establishment of four companies producing wheat that later were dedicated to the extensive agriculture of fruits and vegetables, taking the aquifer to an irreparable overexploitation (*Salameh, Alraggad & Tarawneh, 2014*).

The endorreicity of the Laguna de Santiaguillo basin, its small size, the regulatory vacuum concerning the protection of this type of aquifers in particular and the agriculture that takes place within it are factors that accentuate the vulnerability of the overexploited aquifer. Currently, the aquifer is over-concessioned as a result of poor resource management.

This places agriculture as one of the vital human activities for development that generate greater water stress.

Water quality in an overexploited aquifer is worse than a well-maintained aquifer. A study developed by *Bicalho et al. (2011)* concludes that greater volumes of water extraction and shorter times characterize intensive extraction in aquifers. The mobilization of the deepest water levels increases the mineralization of water while karstification is greater in an overexploited aquifer than in one that is not.

It is quite relevant to note the need for studies that reveal the quality of water in the region since, due to the exploitation of the aquifer, the extracted water presents a greater presence of salts and minerals. In the case of the Laguna de Santiaguillo watershed,

it is imperative to carry out the construction of drainages associated with the tracing of the furrows in contour lines.

Some of the proposals mentioned by *Palacios-Vélez & Escobar-Villagrán (2016)* are applicable for the Santiaguillo aquifer, some of which are the formation of a technical committee for groundwater that is responsible for the management of the aquifer, the artificial recharge of the aquifer and fostering the collection and use of rainwater for domestic and backyard consumption.

The possibility is to guide the technification of irrigation, the control of the livestock population within the basin and the conservation of the vegetation cover of the recharge areas, limiting the expansion of the area devoted to agriculture, the productive conversion of high-yield crops consumption, particularly forages.

However, even if this was possible, it is necessary to consider that the returns of the underground extractions sustain with their contributions the volume of water in the wetland that now has a RAMSAR classification. Therefore, it is necessary to carefully review the implications of such an action as ideal as it may seem.

The balance sheets of this analysis have been made with the official information of the Public Registry of Water Rights of the National Water Commission, in the absence of reliable information from the underground extractions measured. It is not possible to affirm beyond the hypothesis of a condition of physical overexploitation.

According to CONAGUA in its report "Update of the average annual availability of groundwater aquifer (1,001) valley of Santiaguillo" (*Mexican National Commission of Water (CONAGUA), 2015*), the availability of groundwater in the aquifer of Santiaguillo was obtained by subtracting from the volume of total annual recharge, the value of the natural discharge committed and the volume of groundwater concessioned and registered in the REPDA. $-39{,}351{,}828 = 25{,}900{,}000 - 10{,}393{,}000 - 54{,}858{,}828$. This indicates that there is no available volume for new concessions in the hydrogeological unit called Valle de Santiaguillo aquifer, in the State of Durango (*Mexican National Commission of Water (CONAGUA), 2018*).

However, throughout this study it is demonstrated that the problem is even more serious than what was proposed by CONAGUA. Since according to the balance established in this work it was demonstrated that the availability of groundwater is $-111{,}950{,}689.49 = 30{,}023{,}188.87 - 14{,}638{,}517.10 - 127{,}335{,}361.27$, this given that the conagua does not take into account all concessions with title and those of folio 8 or national registration ballot.

In regard to Fig. 7: the status quo represents the current state of the aquifer, which reveals an over-concession of 318%. If we decrease the allowed extraction by 20%, we would be talking about a low reduction of extraction achievable with changes in the cultural practices and basic technifications as well as the reduction of the number of wells through the expiration of rights, even so the extraction would be of 258%, which still places it in a large overexploitation.

Decreasing the extraction concession by 35% is considered an average reduction, which implies a technification of greater scope particularly in irrigation techniques such as the use of central, or lateral pivot systems, sprinkler irrigation, extended use of belt, and multi-damper, this implies an investment in all the uses and even the aquifer would be overexploited by 212%.

The 70% decrease in recorded extraction is actually a high reduction of extraction which implies the advanced technification of irrigation systems such as micro-sprinkling, drip systems, deep cultural change, and changes in crops, which requires a high investment that would necessarily force a government subsidy.

The sub-basins and dams: Guatimape, Tejamen Dam, Tejamen, El Gato, Tinajuelas, Dr. Castillo del Valle Dam, Las Jarillas, El Trébol, Torunos, La Redonda Dam, and El Candado are in balance and feed the northern lagon that has the peculiarity of being a perennial body. On the other hand, the sub-basins Ojo de Lobo, Tinaja, Los monos, El Colorado, and Viborillas feed the southern lagoon which is most dry most of the time (Table 5).

In contrast to the analysis of surface water is the analysis of availability of groundwater (Table 6), the column of infiltration represents the estimated infiltration by rain and it is the most important; however, it is affected by some human activities on the surface such as: agricultural activities and changes in vegetation cover.

The second column presents the returns directly influenced by surface irrigation, only two micro-basins present this type of infiltration: Guatimape and Tejamen, both belonging to the North zone.

The third column expresses the horizontal movements in the aquifer, that is to say, the interaction of water that is found in the subsoil and that generates a dynamic that is affected to a greater or lesser extent by the extractions.

The column of evapotranspiration alludes to the exchange that exists between the subsoil and the atmosphere, the micro-watershed with greater affection by this factor is that of Tinajuelas.

The average annual recharge columns and the natural discharge committed reflect the behavior of the system in a natural way; again the micro-basins of Guatimape and Tejamen turn out to have the highest values in terms of aquifer recharge while having the highest values of natural discharge committed.

The column of underground volume concessioned indicates the volume that is concessioned to the different uses, agricultural, industrial, etc. In regard to underground availability per micro-basin, it is observed that all the micro-basins are in deficit. Only three of them (Ojo de Lobo, Los Monos, and Viborillas) are in positive numbers; it is worth mentioning that the volume concessioned in the latter is minimal

Finally, the exploitation/ recharge column indicates the relationship between extractions and recharges in percentage. In this case, it is important to note the percentage of 783% in the micro-basin of El Gato, where the pressure exerted on it is such that the recharge capacity is substantially surpassed (Table 6; Fig. 7).

It is convenient not to lose sight of the fact that it is a single aquifer, the segmentation obeys to the superficial distribution in micro-watersheds to describe in a regionalized way the pressure that human activities exert on it.

From the comparison of both tables, it is possible to observe that it depends on most of the socio-economic activities of the groundwater, which in terms of agricultural use is five times greater than what the existing surface water infrastructure can provide.

It could be thought that one solution is to promote the use of surface water, but technically this is not possible in all cases, due to the distribution of runoff and the

regionalization of socioeconomic activities. Based on the above, the degree of overexploitation of the underground aquifer is evident, which shows the vulnerability of the system to sustain itself for a long time.

All the rules, issues, and procedures involve peasants, sellers and groups with agriculture interests as well as politicians, legislators, and judges. They control the access to the means of production and establish the economic and social interests in relation to the natural, technological, and human resources. *Moreyra et al. (2012)* mention the relevance of legislation and the entities involved in food production. It is worth mentioning the role of the market since it incentives environmental overexploitation in general to satisfy demand, which most of the time fails to arise from necessity but from overproduction.

In Santiaguillo Basin, there is little efficiency both by the regulator and by society. This apathy may be due to various aspects including the small size of the aquifer as well as the lack of knowledge on the part of some residents regarding the current conditions of overexploitation found in the aquifer. All this contributes to an increase in the vulnerability of the aquifer.

It is, therefore, necessary to determine with sufficient precision the actual volume extracted by direct measurements and monitoring of the volumes delivered by the underground wells, with the purpose of knowing the dynamics and the affectation to specify the degree of vulnerability. With the information generated from this study there is sufficient evidence to place focus on the basin.

In this study, it is evident that there is no way to solve the overexploitation of the aquifer of the Laguna de Santiaguillo basin, but if actions are taken it is possible to mitigate the impacts on the aquifer.

To achieve a good mitigation of the impacts caused to the aquifer, actions are visualized in three sectors: technological actions, cultural practices, and normative aspects.

Technological actions: change the way in which the risks are applied, it is necessary to establish efficient water use system such as drip irrigation, high efficiency systems, sprinkler systems and line, currently the irrigation that is carried out is by flood.

Cultural practices: that is, the way in which the land is worked, level curves, construction of drainage networks.

Regulatory aspects: local agreements that can be built in the *Mexican Underground Water Technical Committee (COTAS) (2018)* (http://www.cocurs.mx/cotas.php, consultation: June/2018), this entity allows to group users working for the improvement and establishment of own regulations. Self-regulations or regulations from an external entity are needed.

## CONCLUSIONS

The population of the Santiaguillo Basin is vulnerable to the exploitation of the aquifer, since all micro-basins remain at the level of water availability per day, it is important to emphasize that agriculture is the economic activity supported by the population and, at the same time, it consumes more than 90% of the water available in the aquifer. Even

considering a scenario where extraction is drastically reduced, the level of water availability for development is catastrophically low, which shows that the remedial actions are unsustainable given the condition of the aquifer. Therefore, it would only be possible to propose mitigation measures to sustain the aquifer and its population for a certain period of time.

Activities such as extended forage production may collapse, thereby threatening farming in the basin.

The micro-basin that has the greatest exploitation is the so-called "El Gato" where there are well-established Mennonite communities, in which the fact of an excessive drilling of well stands out since these communities are immune usually to the application of Mexican regulations on prudent distance of the perforation between well and well. It is extremely important that the regulations permeate all inhabitants of the region, since the actions of a few impact the entire aquifer and affect all the inhabitants of the basin.

An exacerbation of the condition of overexploitation of the aquifer is foreseen, so the measures of control and reduction of the underground water volumes concessioned would have to be a policy established in particular for this area.

Maximizing the use of surface water could contribute, but only very marginally, to reducing overexploitation of the aquifer.

Some of the possible mitigation proposals are: the technification of irrigation, the control of the livestock population within the basin and the conservation of the vegetation cover of the recharge areas, limiting the expansion of the area dedicated to agriculture, the productive conversion of high-consumption crops, particularly forages.

It is necessary to generate governance processes in greater depth, so it is necessary to reach the construction of local agreements between water users to detonate sustainability processes.

However, even if this was possible, it is necessary to consider that the returns of the underground extractions sustain with their contributions the volume of water in the wetland that now has a RAMSAR classification, so it is necessary to carefully review the implications of such an action as ideal as it may seem.

It is therefore necessary to determine with sufficient precision the actual volume extracted by direct measurements and monitoring of the volumes delivered by the underground wells.

It is considered that the results of this research project are sufficient to warn that in this small endorheic sub-basin the productive activities and quality of life are at risk given the vulnerability related to water availability.

### Funding

This work was supported by Instituto Politecnico Nacional (SIP 20151963), Conacyt (Scholarship 408073). The funders had no role in study design, data collection and analysis, decision to publish, or preparation of the manuscript.

## Grant Disclosures

The following grant information was disclosed by the authors:
Instituto Politecnico Nacional: SIP 20151963.
Conacyt: Scholarship 408073.

## Competing Interests

The authors declare that they have no competing interests.

## Author Contributions

- María de Lourdes Corral-Bermudez conceived and designed the experiments, performed the experiments, analyzed the data, contributed reagents/materials/analysis tools, prepared figures and/or tables, authored or reviewed drafts of the paper, approved the final draft.
- Eduardo Sánchez-Ortiz conceived and designed the experiments, performed the experiments, analyzed the data, prepared figures and/or tables, authored or reviewed drafts of the paper, approved the final draft.
- Dioselina Álvarez-Bernal analyzed the data, authored or reviewed drafts of the paper, approved the final draft.
- Martín Omar Gutiérrez-Montenegro conceived and designed the experiments, performed the experiments, analyzed the data, prepared figures and/or tables.
- Erika Cassio-Madrazo analyzed the data, to complete the social vision of the article.

## Data Availability

Complementary maps and a dataset to understand the balance is available at: Raw-Data Availabillity of water Laguna de Santiaguillo Basin, https://figshare.com/projects/Raw-Data_Availabillity_of_water_Laguna_de_Santiaguillo_Basin/60935.

## Supplemental Information

Supplemental information for this article can be found online at http://dx.doi.org/10.7717/peerj.6814#supplemental-information.

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
