# Peer review of "Scenarios of availability of water due to overexploitation of the aquifer in the basin of Laguna de Santiaguillo, Durango, Mexico"

_PeerJ, doi:10.7717/peerj.6814_

## Round 0.1 · original submission · Major Revisions

You did an interesting work and the results can be applied in other areas with similar conditions and problems. But there are some areas which are needed to be improved, particularly, the material and methods section, climate classification, river characteristics, and graphs.

Reviewer 1 ·

Basic reporting

The structure of the manuscript is the classic; there is nothing to say about it.

Experimental design

The method is comprehensible.

Validity of the findings

The article is interesting and could be as a model to transfer to other areas with similar conditions and problems with overexploitation and pollution, the method is comprehensible. This work open new possibilities with the function of the system and the management proposal. It will be interesting that the government consider this work as a part of the management.
Perfect publishable and valid for other researchers.

Additional comments

The aspects that may could be better are the graphics; from my personal point of view are:
The figure 1 must be complete, the orientation and the scale, in the case of the scale if it couldn´t be done there is no problem because in the figure it is marked the surface of the endorheic basin.
The figure 2, in this case the legend is in Spanish and it will be more didactic if the intensity of the color corresponds with the altitude values. In addition, the legend must be organize in equal order, from maximum to minimum altitude.
The limit of the micro-basins must be a line, not a rectangle because it could be associated with a surface.
In the part of “Physical aspects”, I think in the text and not only in the table, it will be fine to specify that the types of clime are the köppen classification.
The hydrography will be more comprehensible and orderly if the figure 6 is added here, and not where is located in the text. This figure has the legend in Spanish and the limits of the basins must be added with a line in the figure. So this will be now figure 3, changing the number of the other figures that follow the text.
The figure 3 is a population pyramid with groups of ages not homogeneous. The legend (man, woman) is in Spanish, but the most important is to add the year. I recommend that the rectangles of the histogram will be designed proportional to the years.
The figure 4 is about uses of land, the legend is in Spanish and to delimit the micro-basins just must appear a line.
The figure 7, the legend is in Spanish and the percentage that it expresses doesn´t specify about what is.
In the article, some phrases are very long; it could be easily to read if there will be short phrases.
In the text, the word sub-basin appear for example in soils, and is the basin.
It is necessary to check all the acronyms, the first time that it appears; it must do it with the complete name of the thing that it represents.

Annotated reviews are not available for download in order to protect the identity of reviewers who chose to remain anonymous.

Reviewer 2 ·

Basic reporting

The structure of the manuscript must be revised
The materials and methods section is not complete.
It is not possible to verify the calculations made since the raw data of the different variables are not known (for example, rain, temperature, evaporation)
The period of time for which the balance was calculated is not identified.
The sources of information on socioeconomic factors are not indicated.
The authors should review the application of the concept of microbasin to the aquifer..
There is no detailed review of the state of the art. There are numerous articles on this subject.
The format of the references must be homogeneous
Do not use the word "underground water". It's groundwater
The legends of the maps are in Spanish.

Experimental design

The materials and methods section is not complete.
Details are displayed in "General comments for the author"

Validity of the findings

Details are displayed in "General comments for the author"

Additional comments

Lines 77-80. The author must use the data of the National Water Commission
(CONAGUA 2017 Water Statistics in Mexico http://sina.conagua.gob.mx/publicaciones/EAM_2017.pdf)
"The annual average 79 rainfall in Mexico is of a volume of 1511 km, which classifies Mexico as a country with low water availability" This sentence should be revised. I recommend using the concept of Degree of Pressure. This concept is found in the previous document.
Lines 91-94. The author must use the data of the National Water Commission
(CONAGUA 2017 Statistics on water in Mexico http://sina.conagua.gob.mx/publicaciones/EAM_2017.pdf)
Line 114. The case study is in Mexico. The authors should include examples of aquifers in the country. These examples should focus on the environmental and economic effects that overexploitation has produced.
Lines 172-173. What type of classification was used? It lacks the reference to the climate classification system. The values of average, maximum and minimum temperature and average precipitation should be included
Lines 240-243. Soil textures. The textural classification of the soil must be more precise. Authors should use the triangle of soil textures (USDA reference)
Line 247. Bibliography reference is not included.
Line 258. The information sources of all these factors (health, education, housing, etc.) must be included. These factors are included in the marginalization index. I would recommend the authors include the values of this index and explain their meaning in detail.
Line 306. The objective of the study includes groundwater. The description of the study area does not include the description of the aquifer and its characteristics.
Line 334. Land use varies over time. The authors must indicate to which year the data correspond.
Line 349 and following. This section is difficult to understand. The authors should bear in mind that non-Mexican readers can not know the method of calculating the balance used in Mexico. This method must be explained in the materials and methods section.
The authors must include raw data of the calculations made to know the availability by microbasin.
Line 419. The balance is established by CONAGUA. The authors can carry out a discussion of their values. For example, buy the recharge data from line 284 with the data presented in this section.
Line 448. This section should go before the surface water balance section
Line 468 and following. The authors should review the application of the concept of microbasin to the aquifer. An aquifer is a geological formation whose limits are those of a hydrogeological nature (impervious materials, faults, etc.). The limits of a basin can not be applied in an aquifer.
Availability calculations with microbasins can only be applied to surface waters.
Table 6. It is not possible to verify if the calculations are correct. Calculations of groundwater balances can not be made by micro-watersheds.
Tables 7, 8, 9, 10 and 11. The method of calculating these indicators is not explained in the materials and methods section.
Figure 5 is repeated with table 4.

Reviewer 3 ·

Basic reporting

There is a mismatch between the title and the content of the paper: the paper only analyses the aspects of overexploitation of the Laguna de Santiaguillo aquifer. I suggest changing the title to this effect.

Related to the aspect mentioned above I suggest replacing your keywords with the following: aquifer overexploitation, fresh water availability, modeling scenarios in watersheds, Laguna de Santiaguillo

The English language needs improvement.

Experimental design

No comment

Validity of the findings

No comment

Additional comments

The paper is not far from a possible publication. Besides the inconsistency between the title and the content (vulnerability of the population being not adequately treated), the work also requires other interventions. Some specific comments are listed in the order of reading the text:

1. the text in lines 71-76 should be supplemented with a more detailed explanation of the term "water availability".
2. lines 89-90: please, replace Banco Mundial with the English name ;
3. lines 172-173: please, give more details about what climate classification is;
4. lines 179-206: it would be more useful to complement the characterization of rivers and their hydrographic basins in terms of flows and hydrological regime;
5. lines 306-309: it is necessary to redefine the aim of the article;
6. lines 335-342: please, explain in more detail how you got the map in Figure 4;
7. the text from the rows 350-353 and 449-452 should be moved to the Methodology section;
8 lines 396, 397, 399, 400, 415, 422, 423 and 425: please express the values in cubic meters (not hectometers);

---

## Round 0.2 · Major Revisions

Please reply to the reviewers' comment one by one, rather than combing all responses together in a single letter. Your current Response Report is not easy to understand and evaluate. It needs to include the actual reviewer comments with a response to each, rather than the more narrative document provided,

Also, please upload both a clean version and a version with change tracks. There are many comments and even two titles in your uploaded ms. It is unacceptable. If you have failed to capture the 'tracked changes' as you were editing, then you could try to use the "Compare documents" tool in Word.

---

## Round 0.3 · accepted · Accept

You have made a good contribution to our understanding about the water over exploitation in Laguna de Santiaguillo, Mexico, using both GIS and field observation.